# Synthesis, Characterization and Anticancer Efficacy Studies of Iridium (III) Polypyridyl Complexes against Colon Cancer HCT116 Cells

**DOI:** 10.3390/molecules27175434

**Published:** 2022-08-25

**Authors:** Biao Xie, Yi Wang, Di Wang, Xingkui Xue, Yuqiang Nie

**Affiliations:** 1The First Affiliated Hospital, Jinan University, Guangzhou 510630, China; 2Department of Gastroenterology, People’s Hospital of Longhua, Shenzhen 518109, China; 3School of Pharmacy, Guangdong Pharmaceutical University, Guangzhou 510006, China; 4Department of Medical Research Center, People’s Hospital of Longhua, Shenzhen 518109, China; 5Department of Gastroenterology, School of Medicine, Guangzhou First People’s Hospital, South China University of Technology, Guangzhou 511458, China

**Keywords:** iridium (III) complexes, apoptosis, cell cycle arrest, autophagy, immunogenic cell death

## Abstract

In this paper, two new iridium (III) complexes, [Ir(ppy)_2_(ipbp)](PF_6_) (**Ir1**) (ppy = 2-phenylpyridine, ipbp = 3-(1*H*-imidazo[4,5-f][1,10]phenanthrolin-2yl)-4*H*-chromen-4-one) and [Ir(bzq)_2_(ipbp)](PF_6_) (**Ir2**) (bzq = benzo[*h*]quinolone), were synthesized and characterized. The cytotoxicity of the complexes against human colon cancer HCT116 and normal LO2 cells was evaluated by the 3-(4,5-dimethylthiazol-2-yl)-2,5-diphenyltetrazolium bromide (MTT) method. The complexes **Ir1** and **Ir2** show high cytotoxic efficacy toward HCT116 cells with a low IC_50_ value of 1.75 ± 0.10 and 6.12 ± 0.2 µM. Interestingly, **Ir1** only kills cancer cells, not normal LO2 cells (IC_50_ > 200 µM). The inhibition of cell proliferation and migration were investigated by multiple tumor spheroid (3D) and wound healing experiments. The cellular uptake was explored under a fluorescence microscope. The intracellular reactive oxygen species (ROS), change of mitochondrial membrane potential, glutathione (GSH) and adenine nucleoside triphosphate (ATP) were studied. Apoptosis and cell cycle arrest were performed by flow cytometry. The results show that the complexes induce early apoptosis and inhibit the cell proliferation at the G0/G1 phase. Additionally, the apoptotic mechanism was researched by Western blot analysis. The results obtained demonstrate that the complexes cause apoptosis in HCT116 cells through ROS-mediated mitochondrial dysfunction and the inhibition of PI3K/AKT signaling pathways.

## 1. Introduction

Cancer, which is a multi-factorial disease featured by the uncontrolled growth of cells in the body, is the second leading cause of deaths worldwide [1]. For many years, cancer has been considered as the result of DNA mutations prompting the activation of oncogenes or the inactivation of tumor suppressor genes [2]. Popovtzer et al. developed a unique method for the specific mechanical lysis of cancer cells using superparamagnetic iron oxide nanoparticle rotation under a weak alternating magnetic field [3,4]. Cisplatin and analogues are extensively used as anticancer drug. However, some inherent limitations still exist in these drugs, such as side effects, general toxicity such as nephrotoxicity, emetogenesis, nerve damage, hair loss, nausea and neurotoxicity and intrinsic acquired resistance [5,6,7,8,9,10,11,12]. These drawbacks in platinum-based anticancer drugs have stimulated more research efforts to investigate drugs based on other transition metals. In the past 20 years, among these metal complexes, the anticancer activity of Ru (II) complexes has been extensively studied [13,14,15,16,17,18]. In recent years, the research in the anticancer activity of iridium (III) complexes has been paid great attention and a number of iridium (III) complexes show potent anticancer efficacy [19,20,21,22,23,24,25,26,27,28]. Yuan et al. reported that two iridium (III) complexes, [Ir(piq)_2_(NPIP)](PF_6_) (piq = 1-phenylisoquinoline, NPIP = 2-(2-nitrophenyl)-1*H*-imidazo[4,5-f][1,10]phenanthroline) and [Ir(bzq)_2_(NPIP)](PF_6_) (bzq = benzo[*h*]quinolone), targeted the mitochondria and the complexes led to the loss of mitochondrial membrane potential (MMP), increased the intracellular ROS content and further induced apoptosis [21]. Complex [Ir(ppy)_2_L]^+^ (ppy = 2-phenylpyridine, L = 1,2-bis(diphenylphosphino)benzene) was specifically targeted to lysosomes in A549 cancer cells. This targeting caused lysosomal damage and the induction of ROS (reactive oxygen species) production in cancer cells [29]. In the reported iridium (III) complexes, the cancer cells are mainly lung cancer, liver cancer and gastric cancer. Few studies reported the efficacy of the iridium (III) complexes against colon cancer. As some chromenone (benzopyranone) derivatives are the subject of considerable pharmaceutical and chemical interest [30,31], it is possible for the complexes containing chromenone to show high anticancer activity. To verify our hypothesis and obtain much information of the effects of the iridium(III) complexes on anticancer activity, in this article, we chose ipbp (ipbp = 3-(1*H*-imidazo[4,5-f][1,10]phenanthrolin-2yl)-4*H*-chromen-4-one) [32] containing chromenone as a ligand to synthesize two new iridium (III) complexes, [Ir(ppy)_2_(ipbp)](PF_6_) (**Ir1**) (ppy = 2-phenylpyridine) and [Ir(bzq)_2_(ipbp)](PF_6_) (**Ir2**) (bzq = benzo[*h*]quinolone). The synthetic route for the complexes is shown in Figure 1. The complexes were characterized by elemental analysis, HRMS, ^1^H NMR and ^13^C NMR. The anticancer activity in vitro was evaluated by the 3-(4,5-dimethylthiazol-2-yl)-2,5-diphenyltetrazolium bromide (MTT) method. As expected, the complexes **Ir1** and **Ir2** show high anticancer activity with a low IC_50_ value of 1.75 ± 0.1 and 6.12 ± 0.2 µM against colon HCT116 cancer cells. The apoptosis, cell cycle arrest, intracellular reactive oxygen species (ROS), the change of mitochondrial membrane potential and the expression of B-cell lymphoma-2 (Bcl-2) family proteins was explored by Western blot analysis. The results show that the complexes induce apoptosis through a ROS-mediated mitochondrial dysfunction and inhibition of the PI3K/AKT signaling pathways, and the complexes may be potent anticancer candidate drugs for colon cancer treatment.

## 2. Experimental

### 2.1. Materials and Method

IrCl_3_·3H_2_O was obtained from Boren Precious Metals Co., Ltd. (Kunming, China). 2-phenylpyridine (Hppy) and benzo[*h*]quinolone (bzq) were purchased from Beijing HWRK Chem Co., Ltd (Beijing). Deionized water was used in all experiments. Elemental analysis was performed by a PerkinElmer 240Q elemental analyzer (USA). For the cellular experiment, Fetal bovine serum (FBS) and Dulbecco’s Modified Eagle’s Medium (DMEM, Gibco, California, USA) purchased from Gibco were used in the cell culture process. Fetal Bovine Serum (FBS, Gibco, California, USA), newborn calf serum (NBCS). Fluorescent probes and assay kits were purchased from Beyotime Biotechnology (Shanghai, China). ^1^H and ^13^C NMR spectra were recorded on a Varian-500 spectrometer (500 MHz) using DMSO-d_6_ as solvent and tetramethylsilane (TMS) as an internal standard at room temperature. The HRMS spectra were measured by direct injection with Waters Xevo G2-XS QTof mass analyzer.

### 2.2. Synthesis of Complexes

#### 2.2.1. Synthesis of [Ir(ppy)_2_(ipbp)](PF_6_) (**Ir1**)

[Ir(ppy)_2_Cl]_2_ [33] (0.232 g, 0.30 mmol) with ipbp (0.218 g, 0.60 mmol) [30] were mixed in a 150 mL of three-neck flask equipped with condenser and stirrer. Then 30 mL of dichloromethane and methanol (*v*/*v* = 2:1) was added into the flask. The mixture was heated and refluxed at 50 °C and stirred for 6 h under argon, a yellow solution was produced. After cooling to room temperature, excess NH_4_PF_6_ powder (0.75 g) was added into the above solution and continuously stirred for 1 h. After the solvent was removed under reducing pressure, a yellow precipitate was obtained. The precipitate was purified through a neutral alumina column with dichloromethane–acetone (*v*/*v* = 2:1) as eluent and a yellow band was collected. Then, the yellow solid was dried in vacuum conditions to obtain a yellow powder. Yield: 75% (0.468 g). Anal. Calc for C_44_H_28_N_6_O_2_IrPF_6_: C, 52.33, H, 2.79, N, 8.32%. Found: C, 52.62, H, 2.56, N, 8.58%. ^1^H NMR (DMSO-d_6_): 9.37 (s, 1H), 8.30 (dd, 1H, *J* = 1.5, *J* = 8.0 Hz), 8.26 (d, 2H, *J* = 8.5 Hz), 8.15 (dd, 2H, *J* = 1.0, *J* = 5.0 Hz), 8.09–8.06 (m, 2H), 7.95 (d, 2H, *J* = 7.5 Hz), 7.89–7.84 (m, 4H), 7.67 (t, 1H, *J* = 7.5 Hz), 7.50 (d, 2H, *J* = 5.5 Hz), 7.06 (t, 2H, *J* = 7.5 Hz), 7.00–6.94 (m, 6H), 6.30 (d, 2H, *J* = 7.5 Hz). ^13^C NMR (DMSO-d_6_, 125 MHz): 176.39, 168.91, 160.48, 157.68, 152.37, 151.15, 150.48, 148.44, 146.24, 146.04, 140.71, 137.17, 133.25, 132.28, 129.03, 128.63, 127.34, 127.07, 125.84, 125.42, 124.41, 121.97, 120.86, 116.65. HRMS (CH_3_CN): *m*/*z* = 866.2824 [(M−PF_6_)^+^].

#### 2.2.2. Synthesis of [Ir(bzq)_2_(ipbp)](PF_6_) (**Ir2**)

This complex was synthesized according to the method described as **Ir1**, with [Ir(bzq)_2_Cl]_2_ [33] in place of [Ir(ppy)_2_Cl]_2_. Yield: 76% (0.496 g). Anal. Calc for C_48_H_28_N_6_O_2_IrPF_6_: C, 54.49, H, 2.67, N, 7.94%. Found: C, 54.25, H, 2.89, N, 8.16%. ^1^H NMR (DMSO-d_6_): 9.36 (s, 1H), 8.52 (d, 2H, *J* = 8.5 Hz), 8.30 (dd, 2H, *J* = 1.5, *J* = 8.0 Hz), 8.13 (d, 2H, *J* = 4.5 Hz), 8.01–7.95 (m, 6H), 7.89 (d, 2H, *J* = 8.5 Hz), 7.85–7.74 (m, 4H), 7.58 (d, 2H, *J* = 6.0 Hz), 7.45–7.43 (m, 2H), 7.22 (t, 2H, *J* = 7.5 Hz), 6.31 (d, 2H, *J* = 7.5 Hz). ^13^C NMR (DMSO-d_6_, 125 MHz): 174.85, 158.92, 158.29, 156.88, 156.12, 149.58, 149.36, 148.66, 147.66, 146.83, 140.84, 138.00, 135.59, 134.22, 130.18, 129.94, 129.00, 127.17, 127.03, 125.82, 124.69, 123.93, 123.23, 120.82, 119.30, 115.27. HRMS (CH_3_CN): *m*/*z* = 913.1222 [(M–PF_6_)^+^].

### 2.3. Cell Culture

HCT-116 (human colon cancer) and human normal liver LO2 cells were purchased from the Tumor Center of Sun Yat-Sen University (Guangzhou, China) and were grown in DMEM culture solution. To provide a suitable environment to sustain cell growth, 100 μg·mL^−1^ streptomycin, 100 U·mL^−1^ penicillin from Beyotime Institute of Biotechnology (Shanghai, China) and 10% FBS were added to the medium. The cell was maintained in the CO_2_ incubator.

### 2.4. Cell Viability Determination

The complexes **Ir1** and **Ir2** were dissolved in DMSO and the final concentration of DMSO was 0.05% in all the cell experiments. The cell viability of HCT116 cells exposure to the different concentration complexes **Ir1** and Ir2 (1.56–100 µM) was performed according to MTT (3-(4,5-dimethylthiazol-2-yl)-2,5-diphenyltetrazolium bromide) method [34]. Briefly, HCT116 cells (4 × 10^3^ cells per well) were placed into 96-well plates and cultured in the incubator (5% CO_2_) at 37 °C for 24 h. Then the cells were treated with different concentrations of the complexes for 48 h, the supernate was removed and 90 μL of serum-free culture medium and 10 μL of MTT (5.0 mg/mL) were added to treat cells for 4 h. The optical density at 490 nm was measured by microplate reader and finally the mean IC_50_ values (three independent experiments) were obtained through SPSS software analysis.

Note: In the following cell experiments, IC_50_ concentration of **Ir1** and **Ir2** are 1.75 and 6.12 μM, respectively.

### 2.5. Cellular Uptake Studies

HCT116 cells were incubated in culture medium at a density of 5 × 10^4^ cells/well in 12-well plates overnight. The cells were treated with IC_50_ concentration of **Ir1** (1.75 μM) and **Ir2** (6.12 μM) and incubated in an incubator at 37 °C and 5% CO_2_ for 4 h, HCT116 cells were stained with DAPI and incubated at 37 °C for 20 min. The cells were photographed under the confocal microscopy (Leica, TCS SP8 SR: Wetzlar, Germany).

### 2.6. Studies of Inhibitory Efficacy in 3D Model

The 3D model was performed according to the literature [21]. In brief, 130 μL of matrigel matrix (10 mg/mL) was added into a pre-chilled confocal Petri dish at 37 °C for 30 min to form gel, then 150 μL of the cell suspension was added into the gel at 37 °C for 30 min. A total of 150 μL of matrigel substrate mixture was slowly added into the confocal Petri dishes to form gel–cell–gel structure and incubated for 7 consecutive days. Then, IC_50_ concentrations of **Ir1** and **Ir2** were added to the confocal Petri dishes for 24 h, the cells were dyed with Live/Dead Cell Kit (Meilum Biotechnology, Dalian) and Hoechst and the cells were washed three times with cool PBS and observed under a confocal microscope (Leica, TCS SP8 SR: Wetzlar, Germany).

### 2.7. Colony-Forming Assay

HCT116 cells in logarithmic phase were trypsinized and placed into a 6-well-plate (200 cells/well). After the cells grow adherently, the IC_50_ concentration of **Ir1** and **Ir2** were used to treat the cells for 24 h. After incubation, the culture solution was replaced with fresh culture solution. Then, the cells were continuously cultured for 10 days and the medium was replaced every 2 days during the colony growth period. After fixing with 70% ethanol and staining using crystal violet (*m*/*v*, 0.1%) for 25 min, they were rinsed twice with PBS and the colony was observed.

### 2.8. Wound Scratch Studies

HCT116 cells (1 × 10^4^ cells /well) were seeded in 6-well plate overnight. After the cell density reached 70%, the pipette tip was used to scratch, then, using phosphate buffer solution (PBS), the wells were washed three times to remove the cell debris. Then the wells were cultured in serum-free medium and incubated with IC_50_ concentration of the complexes at 0 and 48 h. Finally, the cells were photographed under an inverted microscope (Olympus Co.: Tokyo, Japan).

### 2.9. Cell Cycle Arrest Studies

Exponentially growing HCT116 cells were exposed to the IC_50_ concentrations of **Ir1** and **Ir2** for 24 h and the cells were trypsinized. After that, the cells were washed three times with cold phosphate buffer solution (PBS) and fixed with 70% ethanol overnight. Then the cells were dyed with 0.1% Triton X-100 (Biofroxx, Germany), 20 μL of PI (propidium iodide, 0.02 mg/mL) and 20 µL of RNase (ribonuclease, 0.2 mg/mL). Finally, the cell cycle distribution was determined by FACSCalibur flow cytometry (Beckman Dickinson & Co., Franklin Lakes, New Jersey, USA).

### 2.10. Detection of Intracellular Reactive Oxygen Species

2′,7′-dichlorodihydrofluorescein diacetate (DCHF-DA) was utilized to measure the reactive oxygen species (ROS) accumulation. In brief, HCT116 cells (1 × 10^5^ cells/well) were seeded into a blank 6-well plate and maintained in a constant temperature incubator for 24 h, the cells were treated with IC_50_ concentration of **Ir1** and **Ir2** for 24 h and then the cells were dyed with DCHF-DA for 30 min. The cells were washed three times with PBS solution and, finally, the cells were observed and the fluorescence intensity was quantified.

### 2.11. Glutathione (GSH) Measurement

HCT116 cells (5 × 10^5^ cells per well) were seeded in six-well plates overnight. After treatment with IC_50_ or 2IC_50_ concentrations of **Ir1** and **Ir2** for 24 h, the cells were trypsinized and washed three times with PBS. Glutathione (GSH) assay was carried out with GSH Assay Kit (Beyotime Biotechnology: Shanghai, China) according to the manufacturer’s protocol, the absorbance at 412 nm was measured and then the GSH content was calculated.

### 2.12. Mitochondrial Membrane Potential Detection

HCT116 cells were seeded into 12-well-plate (1.2 × 10^5^ cell/well) at 37 °C in the CO_2_ (5%) incubator overnight. The cells were exposed to IC_50_ concentration of **Ir1** and **Ir2** for 24 h, then the cells were washed three times with cool PBS to get rid of the residual complexes. Afterwards, the cells were stained with JC-1 (5,5′,6,6′-tetrachloro-1,1′,3,3′-tetraethylimidacarbocyanine iodide, 1 μg·mL^-1^) for 25 min. Finally, the cells were observed under an ImageXpress Micro XLS system (MD company, San Jose, California, USA).

### 2.13. Assay of ATP Content

HCT116 cells (5 × 10^5^ cell/well) were plated in a 6-well plate overnight. The cells were incubated with IC_50_ and 2IC_50_ concentrations of complexes **Ir1** and **Ir2** for 24 h. The cells were lysed and collected by centrifugation and the supernatant was extracted for subsequent determination. The ATP content of the sample was determined by an enhanced ATP (Adenosine triphosphate) detection kit (Beyotime Biotechnology: Shanghai, China). The concentration of ATP was calculated.

### 2.14. Cell Apoptosis Studies

HCT116 cells were seeded into six-well plates at a density of 5 × 10^5^ cells per well overnight. The cells were cultured in Dulbecco’s Modified Eagle Medium (DMEM, Gibco, California, USA) containing 10% of FBS and incubated at 37 °C (5% CO_2_) for 24 h. The medium was removed and replaced with a new medium containing IC_50_ or 2IC_50_ concentration of **Ir1** and **Ir2** for 24 h. Then HCT116 cells were harvested and washed with PBS three times. The cells were dyed with 50 μg/mL propidium iodide (PI) and 1 mg/mL Annexin V (Beyotime, Sahnghai, China) in the dark for 10 min. The percentage of living, early and late apoptotic cells was determined by a FACS Calibur flow cytometry (Beckman Dickinson & Co., Franklin Lakes, New Jersey, USA).

### 2.15. Expression of Bcl-2 Family Proteins

To quantitatively evaluate the expression of B-cell lymphoma-2 (Bcl-2) family proteins, Western blot was used. After the treatments, the total cell proteins were obtained by lysing HCT116 cells with RIPA (Radio Immunoprecipitation Assay, Cell Signaling Technology, Massachusetts, USA) lysis buffer containing 1% phenylmethanesulfonyl fluoride (PMSF). After centrifugation (13,000 × g, 10 min, 4 °C), the supernatant was gathered. The absorbance values of the protein were measured at 570 nm and diluted to the same concentration with water. Equally denatured protein samples (20 μL protein/lane) were electrophoresed by SDS-PAGE and transferred into polyvinylidene fluoride (PVDF, Millipore: Burlington, MA, USA) membranes. Five percent non-fat milk was used to block the membrane for 3 h in Tris-buffered solution and then maintained at 4 °C overnight. The membranes were then rinsed four times in TBST buffer (150 mM NaCl, 20 mM Tris-HCl, 0.05% Tween 20, pH = 8.0) with shaking and subsequently exposed to conjugated secondary antibodies at 4 °C for 70 min. The residual secondary antibodies were removed by washing with TBST, and each sample was visualized under a computerized imaging system using an enhanced chemiluminescence (ECL) detection kit. Apoptosis-related proteins, including caspase 3, phosphatidylinositol 3-kinase (PI3K), protein kinase B (AKT), Bcl-2-associated x protein (Bax), Bcl-2, PARP (poly ADP-ribose polymerase) and cleaved PARP were quantitatively determined.

### 2.16. Immunostaining Analysis

After HCT116 cells were seeded in 12-well plates and the density reached 70%, the cells were co-incubated with IC_50_ concentration of **Ir1** and **Ir2** for 24 h. The cells were fixed with 70% ethanol and washed with PBS three times, then the immunofluorescence blocking solution was added to block the cells for 1 h. The primary antibody was added and incubated overnight at 4 °C. After that, the fluorescent secondary antibody was added and incubated in dark for 1 h. Finally, the cell nuclei were stained with Hoechst (Beyotime, Shanghai, China) and observed under ImageXpress R Micro XLS System (MD company, San Jose, California, USA).

### 2.17. Data Analysis

All data were expressed as means ± SD. Statistical significance was evaluated by a *t*-test, differences are significant when a * *p* < 0.05, ** *p* < 0.01, *** *p* < 0.001.

## 3. Results and Discussion

### 3.1. Synthesis and Characterization

The ligand ipbp was prepared according to the literature [32]. The complexes **Ir1** and **Ir2** were synthesized by the reaction of [Ir(ppy)_2_Cl]_2_ or [Ir(bzq)_2_Cl]_2_ with ipbp in dichloromethane and methanol. The complexes were purified by column chromatography with dichloromethane/acetone as the eluent and characterized by elemental analysis, HRMS, ^1^H NMR and ^13^C NMR. The determined molecular weight is line with the expected value in the HRMS spectra. In the ^13^C NMR spectra, the peaks of 176.39 ppm for **Ir1** and 174.85 ppm for **Ir2** are attributed to the carbon atom of the carboxyl group. In the ^1^H NMR spectra, the peaks of 9.37 (s, 1H) for **Ir1** and 9.36 (s, 1H) for **Ir2** are assigned to the hydrogen atom H_a_. The peak for the proton on the nitrogen atom of the imidazole ring was not observed. This may be caused by metal coordination inducing electron deficiency in the ligand, therefore, the NH proton of the imidazole ring is very active and easy to be exchanged between the two imidazole nitrogen atoms in the solution [21,35]. The stability of the complexes was determined by the UV-Vis spectra in PBS solution at 0 and 24 h. As shown in Figure 1a, the shape of the peaks has no change, indicating that the complexes are stable in PBS solution. The complexes can luminesce in PBS solution at room temperature (Figure 1b), with a maximum appearing at 624 nm (λ_ex_ = 310 nm) for **Ir1** and 625 nm (λ_ex_ = 310 nm) for **Ir2**, respectively.

### 3.2. Cell Viability and IC_50_ Determination

The ability of the complexes to kill cancer cells was evaluated in terms of cell viability through the 3-(4,5-dimethylthiazole)-2,5-diphenyltetraazolium bromide (MTT) method. As shown in Figure 2a, HCT116 cells were treated with different concentrations (1.56, 3.12, 6.25, 12.5, 25, 50, 100 µM) of **Ir1** and **Ir2** for 48 h and with an increasing amount of the complexes, the cell viability decreased gradually. Therefore, the complexes showed a concentration-dependent inhibition of cell proliferation. The IC_50_ values (concentration of the compound inhibiting cell growth by 50%) were obtained according to the cell viability, and complexes **Ir1** and **Ir2** exhibited very high cytotoxicity toward HCT116 cells with a low IC_50_ value of 1.75 ± 0.10 and 6.12 ± 0.20 µM, respectively. Moreover, **Ir1** showed a stronger ability to kill HCT116 cells than **Ir2**. Comparing the IC_50_ value with the cisplatin (IC_50_ = 15.6 ± 0.4 µM) [36], the complexes show higher cytotoxic activity against HCT116 cells than the cisplatin and gold(III) porphyrin complex (Au(III)porphyrin-adamantane chloride) (IC_50_ = 5.3 µM) [37]. The cytotoxic activity of **Ir1** toward HCT116 cells is comparable with that of the ruthenium metal complex [Ru(phpy)(bpy)_2_]Cl (phpy = 2-phenylpyridine, bpy = 2,2′-bipyridine, IC_50_ = 1.6 ± 0.6 µM) [38]. In addition, the IC_50_ value of **Ir2** toward normal LO2 cells was 40.8 ± 0.3 µM, while the IC_50_ value of **Ir1** against LO2 cells was more than 200 µM, which indicated that complex **Ir1** only kills HCT116 cells, not normal LO2 cells.

### 3.3. Cell Uptake Studies

The prerequisite for the complexes exerting anticancer activity is that the complexes can enter the cells, hence, we first investigated whether the complexes can effectively enter the cancer cells. As shown in Figure 3, during the exposure of HCT116 cells to IC_50_ concentrations of **Ir1** and **Ir2** for 4 h, the cell nuclei were stained blue with 2-(4-amidinophenyl)-6-indolecarbamidine dihydrochloride (DAPI), while the complexes emitted weak green fluorescence. The merge indicates that the complexes can enter the cells and mainly accumulate in the cytoplasm. The results demonstrate that the complexes can be successfully endocytosed by the cells.

### 3.4. Cell 3D Model Studies

Multicellular tumor spheroid cell death was detected using the cell-impermeant indicator Calcein-acetoxymethylester (Calcein-AM) and the propidium iodide (PI) double-staining method. The living cells were stained green with Calcein-AM and the dead cells were stained red with PI. As shown in Figure 4, in the treatment of HCT116 cells with an IC_50_ concentration of **Ir1** and **Ir2** for 24 h, we found that the number of the living cells (green) was reduced and the dead cells (red) increased, compared with that in the control group. The results show that the complexes can effectively inhibit the cell proliferation.

### 3.5. Wound Healing Studies

Metastasis is an intricate process and it involves a series of changes in extracellular matrix digestion, cell migration and invasion [39]. The important role of focal adhesion kinase (FAK) protein in various biological activities is promoting tumor progression and metastasis through targeting the cancer or stromal cells. The ability of the complexes to inhibit cell invasion and metastasis was studied through wound healing experiments. As shown in Figure 5a, the widths reduced in the **Ir1**- or **Ir2**-treated groups compared to those in the control after HCT116 cells were incubated with IC_50_ or 2IC_50_ concentrations of the complexes for 24 h, which reveals that the complexes can inhibit cell invasion. This is also confirmed in Figure 5b where the percentage in the wound healing area decreased in the complex-treated groups compared with that in the control group. Cell colony-forming was also investigated, as shown in Figure 5c where the treatment of HCT116 cells with IC_50_ concentrations of **Ir1** and **Ir2** for 24 h resulted in an obvious decrease in the number of the cells, which indicates that the complexes block cell proliferation. The effect of the complexes on the expression of FAK was explored, as shown in Figure 5d,e, where the HCT116 cells were treated with an IC_50_ or 2IC_50_ concentration of the complexes for 24 h. We discovered that the complexes downregulated the expression of the FAK protein, which further verified that the complexes can effectively inhibit cell invasion and metastasis.

### 3.6. Location and the Change of Mitochondrial Membrane Potential

Mitochondrial changes, including loss of mitochondrial membrane potential, are key events that take place during drug-induced apoptosis [40]. Mitochondria are a critical organelle involved in maintaining cellular homeostasis and producing cellular energy (adenosine triphosphate, ATP) depending on oxidative phosphorylation [41,42]. To investigate whether the complexes enter the cells and then settle at the mitochondria, MitoTracker Red (Beyotime, Shanghai, China) was used as a fluorescence probe. As shown in Figure 6a, in the exposure of HCT116 cells to the IC_50_ concentration of the complexes for 4 h, the mitochondria were stained red and the complexes emitted weak green fluorescence; the overlap of the red and green fluorescence indicates that the complexes enter cells and settle at the mitochondria, which can cause the change in the mitochondrial membrane potential. In the assay of the change in the mitochondrial membrane potential, 5,5′,6,6′-Tetrachloro-1,1′,3,3′-tetraethyl-imidacarbocyanine iodide (JC-1) was used as a fluorescence probe. It is well known that JC-1 forms aggregates and emits red fluorescence corresponding to a high mitochondrial membrane potential. On the other hand, JC-1 exists as a monomer and emits green fluorescence corresponding to low mitochondrial membrane potential. As shown in Figure 6b, in the control, JC-1 emitted bright red fluorescence and very weak green fluorescence, while in carbonylcyanide-m-chlorophenylhydrazone (CCCP, positive control) and the IC_50_ of **Ir1**- and **Ir2**-treated groups, JC-1 emitted bright green fluorescence with little red fluorescence. The changes from the red to green fluorescence demonstrated that the complexes induced a decrease in the mitochondrial membrane potential. Owing to the complexes emitting weak green fluorescence, to exclude the cross-color disturbance, the ratio of red/green fluorescence intensity was determined through flow cytometry with the complexes as a reference. As shown in Figure 6c, compared with the control, the **Ir1** and **Ir2** complexes and cisplatin (CDDP) caused a decrease in the red/green ratio, namely, the red fluorescence intensity decreased and the green fluorescence intensity was enhanced. Taken together, the results showed that the complexes can cause a reduction in the mitochondrial membrane potential.

### 3.7. Intracellular ATP Content Determination

As the most direct source of energy in cells, adenine nucleoside triphosphate (ATP) plays an important role in the cell growth cycle. The interaction of the complexes on the mitochondria will result in a decrease in the intracellular ATP content. As shown in Figure 7, HCT116 cells were treated with IC_50_ and 2IC_50_ of **Ir1** and **Ir2** for 24 h, a decrease in the intracellular ATP content was observed compared with that in the control. Furthermore, the complexes showed a concentration-dependent reduction in the ATP content. Under the same conditions, **Ir1** showed a stronger ability to cause an ATP reduction than **Ir2**, which is consistent with the cytotoxic activity of **Ir1** and **Ir2**. These results reveal that the complexes act on the mitochondria, then induce mitochondrial ATP depletion and further promote the cell death through apoptosis in HCT116 cells.

### 3.8. Intracellular ROS Levels Assay

Many potential anticancer and chemopreventive agents induce apoptosis through the generation of reactive oxygen species (ROS) [43]. The change in the mitochondrial membrane potential is usually accompanied by the change in intracellular reactive oxygen species (ROS). The intracellular ROS level is commonly evaluated using 2′,7′-dichlorodihydrofluorescein diacetate (DCHF-DA) as a fluorescence probe. After the DCHF-DA diffuses inside the cells, it is hydrolyzed by intracellular esterase to yield 2′,7′-dichlorofluorescein (DCFH) [44]. The generated intracellular reactive oxygen species oxidize DCFH to the highly fluorescent compound, namely, 2′,7′-dichlorofluorescein (DCF). The fluorescent intensity of DCF is proportional to the amount of ROS level. As shown in Figure 8a, in the treatment of HCT116 cells with ROSUP (positive control) and the IC_50_ concentrations of **Ir1** and **Ir2** for 24 h, more bright green fluorescence points were found compared with those in the control (weak green fluorescence), indicating that the complexes can enhance the intracellular ROS levels. The DCF fluorescence intensity was quantitatively determined using flow cytometry. To exclude the interference of cross-color fluorescence, the same concentration of the complexes was used as a reference. As shown in Figure 8b, the green fluorescence intensity increased by 1.8 and 3.05 times for the IC_50_ and 2IC_50_ concentrations of **Ir1**, 2.25 and 10.5 times for the IC_50_ and 2IC_50_ concentrations of **Ir2** and 29.7 times for the positive control ROSUP compared with that in the control. The increase in ROS levels followed the order of CCCP > **Ir2** > **Ir1**. This is not in line with the cytotoxic activity of the complexes. These results further confirm that the complexes can increase intracellular ROS levels.

### 3.9. Determination of Glutathione and Lipid Peroxidation

Glutathione (GSH) plays an important role in cellular processes including cancer cell differentiation, proliferation and apoptosis [45]. As shown in Figure 9a, the intracellular concentration of GSH decreased compared with that in the control after a treatment of 24 h of HCT116 cells with the IC_50_ concentration of **Ir1** and **Ir2**. A decrease in the cellular glutathione concentration suggests that the complexes reduce the antioxidant capacity of the cell and finally result in an increase in the reactive oxygen levels and the induction of ferroptosis. Malondialdehyde (MDA) levels in the cell can reflect lipid peroxidation levels. As shown in Figure 9b, HCT116 cells were treated with IC_50_ and 2IC_50_ concentrations of **Ir1** and **Ir2** for 24 h and the intracellular MDA content was enhanced in a concentration-dependent manner compared with that in the control. The results show that the complexes can trigger oxidant stress and further induce ferroptosis.

### 3.10. Apoptotic Efficacy Studies

External aggression by a chemical compound sensed by the cells causes them to undergo two major forms of death, necrosis or apoptosis, each with very distinct characteristics [46]. To evaluate the apoptotic efficacy, the apoptotic percentage in the cell was detected by flow cytometry. As shown in Figure 10a,b, after the treatment of HCT116 cells (I) with an IC_50_ concentration of **Ir1** (II) and **Ir2** (IV) for 24 h, the percentage of apoptotic cells increased by 11.17% for **Ir1** and 6.06% for **Ir2**, respectively, while in the exposure of HCT116 cells to 2IC_50_ concentrations of **Ir1** (III) and **Ir2** (V) for 24 h, the percentage of apoptotic cells was enhanced by 15.57% and 14.47% for **Ir1** and **Ir2**, respectively. The apoptotic efficacy followed the order of **Ir1** > **Ir2**, which is consistent with that of the cytotoxic activity of the complexes. Therefore, the results show that the complexes can effectively induce apoptosis.

### 3.11. Cell Cycle Distribution Studies

The apoptosis of tumor cells is often associated with genomic DNA damage and cell cycle perturbation [7,47]. The distribution of HCT116 cells stained with propidium iodide (PI) in various compartments during the cell cycle was determined using flow cytometry. The DNA distribution histogram of HCT116 cells in the absence and presence of the complexes is shown in Figure 11a. In the control (I), the percentage in the cell in the G0/G1 phase was 40.55%. After HCT116 cells were incubated with IC_50_ concentrations of **Ir1** (II) and **Ir2** (III), an increase in the percentage in the cell in the G0/G1 phase of 10.98% was observed for **Ir1** and 21.17% for **Ir2**, respectively, accompanied by a corresponding reduction of 16.41% for **Ir1** and 26.06% for **Ir2** in the cell in the S phase. The data showed that the complexes inhibited cell proliferation at the G0/G1 phase and the effect of the complexes on the cell cycle arrest followed the order of **Ir2** > **Ir1** under identical experimental conditions. p53, cyclin D1, CDK4 and p21 are related to the cell cycle arrest and regulate the cell cycle in G0/G1. Hence, we detected the expression of these proteins. As shown in Figure 11b,c, the exposure of HCT116 cells to IC_50_ and 2IC_50_ concentrations of **Ir1** and **Ir2** for 24 h resulted in an upregulation of the expression of p53, CDK4 and p21 and a downregulation of the expression of cyclin D1. These results further confirm that the complexes induce cell cycle arrest at the G0/G1 phase.

### 3.12. Autophagy Assay

Cell death includes apoptosis, necrosis and autophagic cell death. Autophagy is an evolutionarily conserved process degrading cellular proteins and cytoplasmic organelles, in which the phagophores surround and pack organelles to form autophagosomes [48]. The Beclin-1 protein is a hallmark of autophagy. As shown in Figure 12a,b, in the incubation of HCT116 cells with the IC_50_ and 2IC_50_ concentrations of **Ir1** and **Ir2** for 24 h, the expression of Beclin-1 increased compared with that of the control. **Ir1** exhibited higher efficacy on autophagy than **Ir2** under the same conditions. Similar results have been observed in other iridium (III) complexes [49,50]. The results show that the complexes can effectively induce autophagy.

### 3.13. The Expression of Bcl-2 Family Proteins

Apoptosis is a kind of programmed cell death regulated by multiple signaling pathways and finally executed by caspase 3 [51]. PI3K (phosphatidylinositol 3-kinase)/AKT (protein kinase B) is an important signaling pathway that causes cell death. Caspase 3 eliminates apoptosis to induce morphological changes. PARP (poly ADP-ribose polymerase) is one of the most important substrates of caspase 3, which is associated with DNA repair and gene integrity [52]. As shown in Figure 13a,b, after the treatment of HCT116 cells with IC_50_ and 2IC_50_ concentrations of **Ir1** and **Ir2** for 24 h, cleaved PARP was found, a decrease in the expression of PI3K, AKT, caspase 3 and B-cell lymphoma-2 (Bcl-2) proteins was observed and the expression of proapoptotic protein Bcl-2-associated x protein (Bax) was increased. The results demonstrate that the complexes can regulate the expression of Bcl-2 family proteins and inhibit the expression of PI3K/AKT. Therefore, the results demonstrate that the complexes induce apoptosis through the inhibition of PI3K/AKT signaling pathway.

### 3.14. Immunogenic Cell Death

Immunogenic cell death (ICD) is a form of cell death. The increase in the cell surface calreticulin (CRT) and high expression of heat-shock protein 70 kDa (HSP70) are typical damage-associated molecular patterns (DAMPs) [53,54]. To evaluate whether the complexes can increase intracellular CRT and HSP70 content, HCT116 cells were exposed to an IC_50_ concentration of **Ir1** and **Ir2** for 24 h and the cells were observed using an ImageXpress R Micro XLS System (MD company, USA). As shown in Figure 14a,b, the green fluorescence intensity was increased, indicating that the complexes can enhance intracellular CTR and HSP70 content. Hence, the complexes can induce immunogenic cell death.

## 4. Conclusions

Two new iridium (III) complexes were synthesized and characterized by elemental analysis, HRMS, ^1^H NMR and ^13^C NMR. The **Ir1** and **Ir2** complexes showed high cytotoxicity against HCT116 cells. The multiple tumor spheroid, colony-forming, wound healing showed that the complexes can effectively inhibit cell proliferation and the cell cycle arrest revealed that the complexes inhibited cell proliferation in the G0/G1 phase. The **Ir1** and **Ir2** complexes can increase the intracellular ROS level, cause autophagy and upregulate the expression of Beclin-1. In addition, the complexes prevent the synthesis of GSH, induce a decrease in the mitochondrial membrane potential and reduce the intracellular ATP content. Western blot analysis showed that the complexes regulate the expression of the Bcl-2 family protein and inhibit the expression of PI3K and AKT. Taken together, the complexes cause apoptosis in HCT116 cells through ROS-mediated mitochondrial dysfunction and the inhibition of PI3K/AKT signaling pathways. This work may help us to understand the anticancer mechanism and synthesis of new iridium (III) complexes as potent candidate drugs for HCT116 cancer treatment.

## Data Availability

All data are available in the manuscript and Appendix A.

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
