# Peer review of "Synthesis, Characterization and Anticancer Efficacy Studies of Iridium (III) Polypyridyl Complexes against Colon Cancer HCT116 Cells"

_molecules, 2022, doi:10.3390/molecules27175434_

Round 1

Reviewer 1 Report

In this paper, the authors introduced in this paper, two new iridium (III) complexes

20 [Ir(ppy)2(ipbp)](PF6) (Ir1) (ppy = 2-phenylpyridine, ipbp = 3-(1H21 imidazo[4,5-f][1,10]phenanthrolin-2yl)-4H-chromen-4-one) and [Ir(bzq)2(ipbp)](PF6) (Ir2) (bzq = benzo[h]quinolone) were synthesized and characterized. The cytotoxicity of the complexes against human colon cancer HCT116 and normal LO2 cells was evaluated by 3-(4,5-dimethylthiazol-2-yl)-2,5-diphenyltetrazolium bromide (MTT) method.The complexes Ir1 and Ir2 show high cytotoxic efficacy toward HCT116 cells with a low IC50 value of 1.75 ± 0.10 and 6.12 ± 0.2 μM. Interestingly, Ir1 only kills cancer cells not normal LO2 cells (IC50 > 200 μM). The idea behind this is interesting. However, I still have quite a number of concerns in this manuscript. There are times where there are not enough data to support the conclusions of the author. Please see some of the major concerns below.

1.The information for the Synthetic route of ligand and complexes Ir1 and Ir2 is not enough. The authors should give much more information about this. So the readers can get its reproducibility. 

2.  The authors should give much more information about the novelty of this paper, especially the effect of using this new iridium complexes , which applications can be used ?

3. More references need to be included in the introduction part to understand the applications for dealing with cancer cells

a.       Targeted magnetic nanoparticles for mechanical lysis of tumor cells by low-amplitude alternating magnetic field

Materials 9 (11), 943

b.       Thermal therapy with magnetic nanoparticles for cell destruction

Biomedical Optics Express 7 (11), 4581-4594

4.  Much more discussion about the results should be given in this paper, especially the author needs to provide enough physicals mechanism analysis about the results.

Author Response

1.The information for the Synthetic route of ligand and complexes Ir1 and Ir2 is not enough. The authors should give much more information about this. So the readers can get its reproducibility. 

 The information for the synthetic route of ligand and complexes has been given in Scheme 1.

  1. The authors should give much more information about the novelty of this paper, especially the effect of usingthis new iridium complexes , which applications can be used ?

 The effect and application have been added in the introduction.

  1. More references need to be included in the introduction part to understand the applications for dealing with cancer cells
  2. Targeted magnetic nanoparticles for mechanical lysis of tumor cells by low-amplitude alternating magnetic field, Materials 9 (11), 943
  3. Thermal therapy with magnetic nanoparticles for cell destruction, Biomedical Optics Express 7 (11), 4581-4594

These literatures have been cited in the introduction.

  1. Much more discussion about the results should be given in this paper, especially the author needs to provide enough physicals mechanism analysis about the results.

The discussion has been revised.

Reviewer 2 Report

In this manuscript ‘Synthesis, characterization and anticancer efficacy 3 studies of iridium (III) polypyridyl complexes 4 against colon cancer HCT116 cells’ author need to explain many things before accepting and comments are below………

1) Introduction section: The necessity and innovation of the article should be presented to the introduction.

2) In the results section: Author perform MTT assay to check viability of HCT116 cell at different concentration as shown figure 2a. However, I am not able see the cytotoxicity level of the normal cell at different concentration of Iridium complex. The figure quality is too bad and I am not able to see properly. Moreover, the another issue I noticed the cell number is very low to check cell viability and the ideal number of cells at least 50000 cell. Author must be explain properly and give the data or perform MTT assay to show the cytotoxicity effect of these iridium complex in the normal cell line. Author must be show the statistical significance between control and experiment part.

3) For cellular uptake study: Author must be explaining which IC50 concentration of Ir1 and Ir2 taken. There is no any clear written in the result section and material methods section. Author also needs to justify why author taken different cells number for different assay? Second question why author choose only 4 hour to identify the cellular uptake assay why not choose some other time point also such as 2h, 4h and 6h.

4)  Figure 5b: I want to see original gel picture and author must be submit original gel picture in the supplementary figure.

5) The flow cytometry data is not clear and author must be revise by adding isotype control and I see the figure there is no any different found between all these control vs. experiment. However, I saw the number of cell density is higher in Ir2 but number written is lower (84.35) compared to control group.

6) Figure quality is very bad author need to revise all figure at high resolution.

7) It is suggested to compare the results of the present research with some similar studies which is done before.

8) Discussion is not written properly author must be rewrite the discussion with concise way

9) It is suggested to organize Conclusion section much better. This section should present in one 250-300 words paragraph.

10)  There are lot of punctuation and typographical errors throughout in the manuscript. Unfortunately, I can’t correct throughout. It must be rechecked by native English speaker.

Author Response

In this manuscript ‘Synthesis, characterization and anticancer efficacy 3 studies of iridium (III) polypyridyl complexes 4 against colon cancer HCT116 cells’ author need to explain many things before accepting and comments are below………

1) Introduction section: The necessity and innovation of the article should be presented to the introduction.

 The necessity and innovation have been added in the introduction.

2) In the results section: Author perform MTT assay to check viability of HCT116 cell at different concentration as shown figure 2a. However, I am not able see the cytotoxicity level of the normal cell at different concentration of Iridium complex. The figure quality is too bad and I am not able to see properly. Moreover, the another issue I noticed the cell number is very low to check cell viability and the ideal number of cells at least 50000 cell. Author must be explain properly and give the data or perform MTT assay to show the cytotoxicity effect of these iridium complex in the normal cell line. Author must be show the statistical significance between control and experiment part.

The figure for IC50 values has been deleted, and the cell viability for normal LO2 is depicted in Figure 1b.

Figure 1a has been revised. In the beginning, the cell number is 4000, after the cells were seeded in the plates for 24 h, the cell density reaches 70-80%, then the cells were treated with different concentration of the complexes for 48 h, finally, the cell numbers are more than 50000. The IC50values of the complexes against normal LO2 have been obtained. The statistical significance has been added.

3) (1) For cellular uptake study: Author must be explaining which IC50 concentration of Ir1 and Ir2 taken. There is no any clear written in the result section and material methods section. (2) Author also needs to justify why author taken different cells number for different assay?

(1) In all cell experiments, the concentrations of the complexes are IC50 concentration, namely, the concentrations for Ir1 and Ir2 are1.75 and 6.12 μM. This has been added in section 2.4 cell viability determination.

(2) In the different assays, in the beginning, the number of the cells are different. When the density of the cells reaches 70-80%, the cells were treated with the complexes for 24 h, finally, the cell numbers are the same.

Second question why author choose only 4 hour to identify the cellular uptake assay why not choose some other time point also such as 2 h, 4 h and 6 h.

 In our previous work, we found that the complexes can enter to the cells after 4 h, hence, we performed the cellular uptake to choose 4 h.

4)  Figure 5d: I want to see original gel picture and author must be submit original gel picture in the supplementary figure.

The gel pictures for figure 5d, we provide all gel pictures in the supplementary information

   FAK               actin

5) The flow cytometry data is not clear and author must be revise by adding isotype control and I see the figure there is no any different found between all these control vs. experiment. However, I saw the number of cell density is higher in Ir2 but number written is lower (84.35) compared to control group.

This experiment has been performed again, and Figure 6c has been replaced.

6) Figure quality is very bad author need to revise all figure at high resolution.

 The figures have been revised.

Round 2

Reviewer 1 Report

The new version can be published.

Reviewer 2 Report

Still some figure quality is not good and author must be improved.